# A Closer Look at In-Distribution vs. Out-of-Distribution Accuracy for Open-Set Test-time Adaptation

**Zefeng Li**  *zefengli@cs.ubc.ca*
*University of British Columbia and Vector Institute*

**Evan Shelhamer**  *shelhamer@cs.ubc.ca*
*University of British Columbia and Vector Institute*

**Reviewed on OpenReview:** *https://openreview.net/forum?id=4MuLx2YDmi*

## Abstract

Open-set test-time adaptation (TTA) updates models on new data in the presence of input shifts and unknown output classes. While recent methods have made progress on improving in-distribution (InD) accuracy for known classes, their ability to accurately detect out-of-distribution (OOD) unknown classes remains underexplored. We benchmark robust and open-set TTA methods (SAR, OSTTA, UniEnt, and SoTTA) on the standard corruption benchmarks of CIFAR-10-C (Krizhevsky, 2009; Hendrycks & Dietterich, 2019) at the small scale and ImageNet-C (Deng et al., 2009; Russakovsky et al., 2015; Hendrycks & Dietterich, 2019) at the large scale. For CIFAR-10-C, we use OOD data from SVHN (Netzer et al., 2011) and CIFAR-100 (Krizhevsky, 2009) in their corrupted (-C) forms. For ImageNet-C, we use OOD data from ImageNet-O (Hendrycks et al., 2021b) and Textures (Cimpoi et al., 2014) in likewise corrupted (-C) forms. ImageNet-O is more similar to ImageNet, with new but related object classes (like "garlic bread" vs. "hot dog" for food, or "highway" vs. "dam" for infrastructure), while Textures is more different from ImageNet, with non-object patterns (like "cracked" mud, "porous" sponge, "veined" leaves). We evaluate the accuracy and confidence of TTA methods for InD vs. OOD recognition on CIFAR-10-C and ImageNet-C with multiple metrics. We also verify the accuracy of each method's own OOD detection technique on CIFAR-10-C. We further examine more realistic settings, in which the proportions and rates of OOD data can vary. To explore the trade-off between InD recognition and OOD rejection, we propose a new baseline that replaces softmax/multi-class output with sigmoid/multi-label output. Our analysis shows for the first time that current open-set TTA methods struggle to balance InD and OOD accuracy and that they only imperfectly filter OOD data for their own adaptation updates.

## 1 Introduction

Deep networks can achieve excellent accuracy on training and test data from the same distribution. However, their accuracy often degrades when the test data differs from the training data due to distribution *shift* (Quionero-Candela et al., 2009). This is a challenge for deployment, as the data may include both shifted inputs of known (InD) classes and unknown (OOD) classes, which requires both generalization to shift for the known InD classes and rejection of the unknown OOD classes. Test-time adaptation/training (TTA/TTT) methods mitigate shift by updating on the test data during deployment (Schneider et al., 2020; Sun et al., 2020; Wang et al., 2021). Their extensions to open-set TTA try to update on InD data from known classes while filtering out unreliable or OOD data from unknown classes (Niu et al., 2023; Lee et al., 2023). While such methods improve InD accuracy, their OOD accuracy for the correct rejection of unknown classes has not been closely examined.

However there is a practical need for recognizing and rejecting OOD data. Mistaking OOD for InD could have costly repercussions, for instance for robotics or for an online moderation system, where actions could be difficult to reverse or require human effort to correct. For this reason, existing adaptation methods either apply robust optimization techniques (Niu et al., 2023), or try to filter InD data from OOD data to only update on the InD or apply a different update on the OOD (Gao et al., 2024; Gong et al., 2023; Lee et al., 2023).

While these methods evaluate InD accuracy, and sometimes evaluate the OOD metric of open set classification rate (OSCR) (Dhamija et al., 2018), existing evaluations do not fully consider OOD performance, nor do they consistently evaluate it across multiple settings and metrics. This is a critical gap, because failure to recognize OOD data could result in a high rate of false positives, and incur costly corrections or incorrect downstream decisions. We identify this gap, experiment to provide these missing measurements for established methods, and gauge their performance with a new baseline that reparameterizes the classification predictions. We find that current methods do not effectively balance InD and OOD accuracy, and furthermore we identify missing conditions for different combinations of InD and OOD data. The current open-set setting typically assumes that each batch contains the same proportion of InD and OOD data. However, this assumption is not realistic. In practice, the OOD proportion within a batch may vary over time, and in extreme cases, there may be entire OOD batches. We therefore evaluate under these broader settings and find that the OOD proportion has a stronger impact on existing methods. In addition, we investigate how different normalization layers affect the stability of test-time updates on open-set data, as a counterpart to this investigation on closed-set data (Niu et al., 2023). In summary, we provide a more complete empirical understanding of open-set test-time adaptation with standardized metrics, a new baseline that strikes a different InD/OOD trade-off, and new benchmark settings for the combination of InD/OOD data in varying amounts.

## 2 Related Work

Domain adaptation (DA) aims to transfer knowledge from a labeled source domain to a target domain where distribution shifts occur (Quionero-Candela et al., 2009). Unsupervised domain adaptation (UDA) specifically addresses fully unlabeled target domains (Saenko et al., 2010; Ganin & Lempitsky, 2015). Early UDA methods typically assumed identical categories between source and target domains, known as the closed-set condition. However, this assumption may not hold in practical applications. Consequently, more realistic scenarios have emerged: open-set settings, where the target may contain unknown categories (Panareda Busto & Gall, 2017; Saito et al., 2018), and partial-set settings, where the target label space is a subset of the source, and further settings with unknown source-target category relationships (You et al., 2019). These scenarios are more difficult and have inspired more flexible and robust adaptation techniques. However, they all still assume joint access to source and target data, and without any bound on the computation needed, so practical deployment may remain difficult if the data or computation are limited.

Test-time adaptation (TTA) and test-time training (TTT) have recently emerged as alternatives for handling shifts without access to the target data during training. Unlike prior domain adaptation methods, which requires target data before deployment, TTA/TTT methods update the model online using only test inputs from the target domain(s). The earliest method focused on straightforward modifications of batch normalization (BN) statistics (Schneider et al., 2020), where the running mean and variance are updated using target data to alleviate distribution mismatch. Tent (Wang et al., 2021) minimizes entropy and performs model updates with respect to the normalization affine parameters during inference, which enables the model to become more confident on target inputs, and results in more correct predictions. Test-time training (TTT) (Sun et al., 2020), in contrast, augments the model with an auxiliary self-supervised task learned during training, and then at inference time the model optimizes only the auxiliary task loss on each test input, allowing it to adapt even when no task labels are available. These approaches demonstrate the feasibility of online adaptation under stricter test-time constraints on data and computation. However, generalization to broader and more complex shifts is still challenging, especially when there is shift in the labels alongside shift in the data.

Open-set TTA extends from shifted data to shifted labels: testing now includes inputs from unknown classes not seen during training. In this setting, adaptation must simultaneously update on inputs from

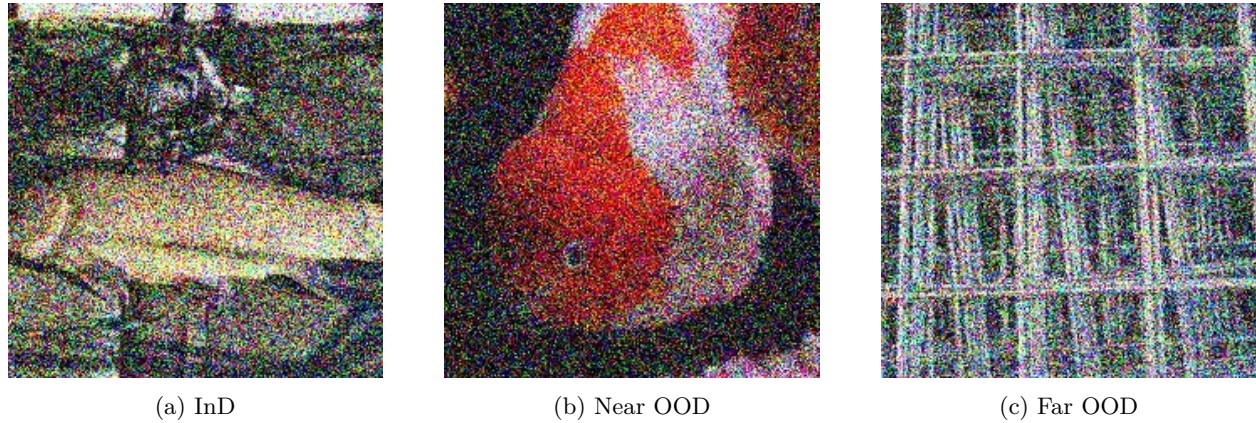

(a) InD                          (b) Near OOD                         (c) Far OOD

Figure 1: Example images of InD (ImageNet-C), Near OOD (ImageNet-O-C), and Far OOD (Texture-C) data corrupted by Gaussian noise. We study adaptation to corruptions on InD and OOD data.

known classes and reject or otherwise handle inputs from unknown classes without supervision of known classes or InD vs. OOD. Most existing open-set TTA methods specialize to this challenge by filtering (estimated) InD and OOD inputs. SoTTA (Gong et al., 2023) filters by confidence of prediction, only high-confidence predictions are added to the memory for adaptation. OSTTA (Lee et al., 2023) filters by aggregate statistics, where low-confidence inputs are identified and gradually downweighted across updates, by relying on predictions across steps ("group wisdom"). UniEnt(+) (Gao et al., 2024) minimizes entropy for inputs filtered as InD and maximizes entropy for inputs filtered as OOD.SAR (Niu et al., 2023), while not an open-set TTA method, is a robust TTA method that includes filtering. It first rejects high-entropy inputs, then rejects low-entropy inputs with unstable gradients, in order to improve the reliability of adaptation with small batch sizes, multiple shifts, and differing proportions of categories. Although each method has been evaluated in a variety of experiments, there has not been a standardized evaluation nor a close examination of their filtering.

While these methods improve accuracy on InD data, they have only been thoroughly evaluated by this metric, without as much attention to their performance on OOD data. Their accuracy on OOD data, that is the rate of correct rejection on data from classes not seen during training, is a missing metric in many cases: without it our knowledge of open-set TTA performance is incomplete. Our benchmarking and analyses complete the picture, with a more consistent evaluation of InD and OOD metrics across current methods and simple baselines, to inform further progress on open-set methods.

## 3 Benchmarking the InD and OOD Accuracy of Test-Time Adaptation

We benchmark InD and OOD accuracy together, to measure current performance and examine their relationship, for a fuller understanding of the practicality of open-set TTA. Existing open-set methods address the filtering of OOD data, and can make use of confidence measures like entropy, but our purpose is to more systematically examine their performance. We gauge the effect of their filtering and updates on both InD accuracy and OOD metrics and in comparison to simpler baselines that threshold confidence alone for different output distributions.

### 3.1 Benchmark Setup

We evaluate open-set accuracy by combining a pair of InD and OOD datasets following prior experiments (Gao et al., 2024). We mix the datasets so each test batch is 50% InD and 50% OOD. Each TTA method adapts and predicts on this mixed dataset, and we measure the accuracy for the known/InD and unknown/OOD data over the full test set. To summarize we report their average or "mixed" accuracy.

**Data.** We evaluate adaptation to image corruptions across multiple datasets (see Figure 1 for examples). CIFAR-10-C (Hendrycks & Dietterich, 2019) is a corrupted version of CIFAR-10 (Krizhevsky, 2009), which

serves as InD test data, because it is a standard dataset for TTA. SVHN-C (Netzer et al., 2011) is a version of Street View House Numbers with the same corruptions. SVHN-C serves as an OOD test data with unknown classes: the SVHN classes, the numbers $\{0, \dots, 9\}$, have no intersection with the CIFAR-10(-C) classes. This setup has been widely adopted in prior work on open-set adaptation (Lee et al., 2023; Gao et al., 2024). CIFAR-100-C (Hendrycks & Dietterich, 2019) is a version of CIFAR-100 (Krizhevsky, 2009) with the same corruptions. CIFAR-100-C serves as alternative OOD test data to assess the effect of OOD data that is more similar (CIFAR-100-C) or less similar (SVHN-C).

We also experiment at the larger scale of ImageNet-C(Hendrycks & Dietterich, 2019) and ImageNet-O-C(Hendrycks et al., 2021b). ImageNet-C is a version of ImageNet(Deng et al., 2009) with the same corruptions. ImageNet-O(-C) consists of different classes that are not in ImageNet, but are visually similar to classes that are in ImageNet, and is used to evaluate open-set recognition performance. The Texture(-C)(Cimpoi et al., 2014) dataset contains texture-centric images with rich appearance patterns but no semantic objects. We use it as an additional OOD test set for ImageNet-C. Each dataset contains 5 severity levels, and we use the most severe corruption level of 5, following established practice (Wang et al., 2021).

The evaluation of online adaptation requires batching and ordering the data. We use a batch size of 200 for CIFAR-10-C experiments following Gao et al. (2024) and 64 for ImageNet-C experiments following Wang et al. (2021). We evaluate the standard sequence of 15 corruption types (Hendrycks & Dietterich, 2019; Niu et al., 2023). On CIFAR-10-C we evaluate episodic adaptation over shifts by resetting the model between each corruption (Wang et al., 2021). On ImageNet-C we evaluate continual adaptation over shifts by not resetting the model (Niu et al., 2023; Lee et al., 2023; Gao et al., 2024).

**Models and Methods.** For CIFAR-10-C, we choose the Wide ResNet-28-10 architecture (Zagoruyko & Komodakis, 2016) with the `Hendrycks2020AugMix_WRN` parameters (Hendrycks et al., 2020) from RobustBench (Croce et al., 2021). For ImageNet-C, we use the standard ResNet-50 (BN) model (He et al., 2015).

We evaluate SAR, as a robust method, and SoTTA, OSTTA, and UniEnt(+) with Tent as open-set methods. These methods require a base TTA method for extension, and so we choose Tent for comparability with our baseline. We use Tent as a simple baseline without customization for robustness or open-set adaptation. SAR is designed for specific normalization layers (GN/LN), and use of BN is discouraged (Niu et al., 2023). However, to ensure consistency and fair comparability across methods, we evaluate SAR using the same model with BN in our main experiments. In addition, to also report SAR in its intended configuration, we evaluate SAR with ResNet-50 (GN) and ViT-B/16 (LN) on ImageNet.

**Metrics.** We measure accuracy (%) on InD and OOD data. For InD inputs, a prediction is counted as correct if the model assigns the correct class label; for OOD inputs, a prediction is counted as correct if the model rejects it. We also measure the "mixed" accuracy that averages the InD and OOD accuracies, providing an overall measure of performance. For a more thorough evaluation of OOD performance, we report the area under the receiver operating characteristic curve (AUROC), the false positive rate at 95% true positive rate (FPR@TPR95), and the open-set classification rate (OSCR)(Dhamija et al., 2018).

### 3.2 Sigmoid Output Instead of Softmax Output

Existing open-set TTA methods are based on softmax output, which forces the prediction of a class for every input. This constraint prevents the model from rejecting unknown inputs, by predicting no class, making it difficult to separate InD from OOD inputs. In contrast, sigmoid output produces independent probabilities for each class, allowing all outputs to remain low simultaneously. This property enables explicit rejection when the model's confidence is insufficient, making the sigmoid a potentially more suitable choice for open-set scenarios. To study how different output distributions influence InD–OOD separation, we replace the softmax nonlinearity with the sigmoid nonlinearity for each class output and update it using sigmoid cross-entropy loss during test-time adaptation.

To ensure that the output distribution and entropy used in testing are compatible, we fine-tune the model on the source data by updating only the final linear layer. We control for accuracy by ensuring the fine-tuned sigmoid model achieves similar accuracy on the standard training and test set as the original softmax model to avoid potential confounds from fine-tuning. On CIFAR-10, the original softmax model achieves 93.91%

Table 1: Accuracy and OOD metrics for TTA on CIFAR-10-C (InD) and SVHN-C (OOD).

| Method | Acc↑ | AUROC↑ | FPR↓ | OSCR↑ |
|---|---|---|---|---|
| Source | 81.74 | 77.89 | 79.46 | 68.44 |
| Source(Sigmoid) | 81.49 | 80.63 | 75.14 | 70.98 |
| Tent | 85.26 | 70.02 | 82.73 | 63.74 |
| Tent(Sigmoid) | 85.08 | 83.01 | 72.06 | 74.65 |
| UniEnt(Tent) | 84.64 | 87.22 | 61.34 | 77.44 |
| UniEnt+(Tent) | 84.45 | 87.43 | 60.70 | 77.45 |
| OSTTA | 85.13 | 74.48 | 79.86 | 67.50 |
| SoTTA | 86.28 | 68.13 | 88.76 | 63.68 |
| SAR(BN) | 85.46 | 80.91 | 75.17 | 73.29 |
| SAR(BN+Sigmoid) | 84.92 | 83.41 | 70.34 | 75.32 |

accuracy, while our fine-tuned model achieves 93.90%. In contrast, directly replacing the final softmax layer with a sigmoid without fine-tuning leads to a significant drop to 89.28% accuracy, indicating that the fine-tuning step is necessary to restore the model's performance and ensure a fair comparison starting from similar clean accuracy.

To evaluate the effect of adopting sigmoid output across methods, we integrate sigmoid with two closed-set TTA methods: Tent and SAR. Tent performs test-time adaptation by minimizing the entropy of predictions, whereas SAR further incorporates filtering based on entropy and gradient stability to exclude unreliable updates. The switch to sigmoid output has little effect on the computation for adaptation. On ImageNet-C, our timing results show that the original Tent and Tent (sigmoid) both take about 175 ms per batch, so the sigmoid replacement does not add noticeable runtime overhead. The peak GPU memory increases slightly from 5533.2 MB to 5633.2 MB (about 100 MB, $<2\%$) for only marginal memory overhead.

### 3.3 Confidence Analysis of InD vs. OOD

We analyze recognition of InD vs. OOD by confidence to standardize comparison across methods. We score the confidence of the output probabilities $\mathbf{p}$ by (1) entropy

$$H(\mathbf{p}) = -\sum_{i=1}^{C} p_i \log p_i, \text{ or} \tag{1}$$

(2) maximum probability or softmax response (SR) (Geifman & El-Yaniv, 2017)

$$SR(\mathbf{p}) = \max_i p_i. \tag{2}$$

For ease of interpretation, we normalize the entropy to $[0, 1]$, and we instead measure $1 - \text{SR}$, so higher confidence is indicated by lower values for both scores across the same range.

These measures are used to classify an input as InD or OOD by thresholding. Given the predictions for the full test, we sweep the threshold to measure the trade-off between InD recognition and OOD rejection.

## 4 Results for InD, OOD, and Mixed Accuracy

We report the accuracy on InD and OOD data as a function of the confidence score (entropy or 1 - softmax response) and threshold. For each experiment we tune the threshold to maximize the mixed accuracy to balance InD and OOD accuracy. The mixed accuracy is simply the mean of the InD and OOD accuracies.

For tuning the hyperparameters, including the thresholding of the confidence scores that is a focus of our study, we follow two established schemes for test-time adaptation. The first scheme is optimistic: we tune to the test shifts to find one shared setting of the hyperparameters for the fifteen types of corruptions. The second scheme is realistic: we tune to the held-out shifts (speckle noise, Gaussian blur, spatter, and saturate)

Table 2: Accuracy on CIFAR-10-C (InD) and SVHN (OOD) with entropy confidence. The results are with the best threshold for mixed accuracy (values in parentheses are with the best threshold for InD accuracy).

| Method | InD Acc ↑ | OOD Acc ↑ | Mixed Acc ↑ |
|---|---|---|---|
| Tent | 70.19(85.26) | 59.79 | 64.99 |
| Tent(Sigmoid) | 62.12(85.08) | 60.70 | 61.41 |
| UniEnt(Tent) | 65.22(84.65) | 89.54 | 77.38 |
| UniEnt+(Tent) | 64.87(84.45) | 90.08 | 77.48 |
| OSTTA | 69.43(85.10) | 65.24 | 67.34 |
| SoTTA | 72.98(86.15) | 43.97 | 58.48 |
| SAR(BN) | 66.08(85.49) | 78.50 | 72.29 |
| SAR(BN+Sigmoid) | 65.57(84.92) | 79.89 | 72.74 |

Table 3: Accuracy of open-set TTA on CIFAR-10-C (InD) and SVHN-C (OOD) with max. prob. confidence.

| Method | InD Acc ↑ | OOD Acc ↑ | Mixed Acc ↑ |
|---|---|---|---|
| Tent | 74.81(85.27) | 49.22 | 62.02 |
| Tent(Sigmoid) | 69.88(85.03) | 79.67 | 74.78 |
| UniEnt(Tent) | 70.84(84.67) | 84.08 | 77.46 |
| UniEnt+(Tent) | 70.53(84.44) | 81.96 | 76.25 |
| OSTTA | 74.20(85.11) | 53.90 | 64.05 |
| SoTTA | 76.63(86.26) | 36.84 | 56.74 |
| SAR(BN) | 72.25(85.49) | 66.51 | 69.38 |
| SAR(BN+Sigmoid) | 69.82(84.92) | 72.82 | 71.32 |

then apply these shared hyperparameters to all of the test shifts (Rusak et al., 2022). In our experiments both tunings select the same thresholds to achieve the highest in-distribution accuracy or mixed accuracy per the choice of tuning objective. Their agreement means that these results simultaneously serve as an analysis of oracle results, with the best thresholds for the test shifts, and as an evaluation of deployable methods, with the best thresholds for the held-out shifts.

Note that UniEnt and SAR require design choices for our experiments. UniEnt is designed to extend a base TTA method, so we combine it with Tent for simplicity and comparability. SAR is designed for models with normalization by GN or LN; however, for consistency across methods, we apply it to the same model with BN. These controlled experiments may not reflect the full potential of SAR with other models. Because the range and distribution of the sigmoid entropy differs substantially from those of the softmax entropy, we adjust the SAR hyperparameters to compensate. We tune its learning rate and the entropy threshold and report the best-performing configurations for CIFAR-10-C and ImageNet-C. Nevertheless we mark these results in grey. To reflect the full performance of standard SAR, we also evaluate SAR with GN on ImageNet-C in Tab. 10 (following the paper and its use of ResNet-50 with GN), and we more fully examine the role of normalization layers for open-set SAR across ResNet-50 and ViT-B/16 models in Sec. 4.9.

## 4.1 Accuracy using Different Confidence Scores

Table 1 reports performance on CIFAR-10-C and SVHN-C using accuracy and three OOD detection metrics: AUROC, FPR@TPR95, and OSCR. The different methods achieve comparable InD accuracy, while UniEnt shows the best separation between InD and OOD. Our proposed Tent (Sigmoid) and SAR (Sigmoid) slightly lag behind the original methods in InD accuracy, but yield a clear improvement in OOD separation.

Tables 2 (entropy) and 3 (max probability) report the results when selecting the best threshold for mixed accuracy. For completeness, and to emphasize the existing default of tuning for InD accuracy, we also report the top InD accuracy across thresholds in parentheses. There is a clear trade-off: achieving the highest

Table 4: Accuracy of open-set TTA methods on CIFAR-10-C (ID) and SVHN (OOD) with their own filtering.

| Method | InD Acc ↑ | OOD Acc ↑ | Mixed Acc ↑ |
|---|---|---|---|
| Tent | 85.26 | 0.00 | 42.63 |
| UniEnt(Tent) | 68.50 | 77.04 | 72.77 |
| UniEnt+(Tent) | 70.60 | 73.47 | 72.04 |
| OSTTA | 51.98 | 52.61 | 52.30 |
| SoTTA | 61.39 | 68.02 | 64.71 |
| SAR(BN) | 81.94 | 8.96 | 45.45 |
| SAR(BN+Sigmoid) | 83.52 | 5.98 | 44.75 |

mixed accuracy, and improving the OOD accuracy, comes at the cost of a significant drop in InD accuracy. The trade-off is sharp in the opposite direction: selecting the threshold to maximize InD accuracy results in OOD accuracy near zero. In general, entropy achieves higher mixed accuracy than max probability.

The state-of-the-art UniEnt achieves the highest mixed accuracy. Our new baseline of Tent (Sigmoid), which replaces softmax output with sigmoid output, is second best for max probability confidence. It achieves this mixed accuracy by exchanging InD and OOD accuracy: its InD accuracy is lower than Tent, but its OOD accuracy is significantly higher. Note that Tent (Sigmoid) lacks any explicit technique for OOD detection, unlike the state-of-the-art UniEnt/UniEnt+, but the choice of output distribution has its own effect.

## 4.2 Checking OOD Filtering Techniques in TTA

We evaluate how well robust and open-set TTA methods filter OOD data.

- **Tent** *does not filter* and updates on all inputs.

- **SAR** filters inputs that exceed a fixed threshold on entropy.

- **UniEnt** scores each input as InD/ODD based on its similarity to class prototypes, clustering by GMM with hard assignment, and filters those assigned to the cluster for OOD data.

- **UniEnt+** estimates the probability of InD vs. OOD by learned scores and filters inputs when $P_{\text{InD}} - \alpha \cdot P_{\text{OOD}} < 0$.

- **OSTTA** filters inputs based on confidence improvement: an input is filtered if confidence *drops* after its update.

- **SoTTA** filters inputs by thresholding the softmax response.

Table 4 shows that method-specific techniques for recognizing InD vs. OOD inputs are not perfect. Their accuracy can even be lower than the best threshold in our confidence analysis. This underlines the focus of current methods on InD accuracy, without necessarily ensuring the reliability of OOD detection, and the opportunity to improve filtering.

## 4.3 Distribution of Different Confidence Scores

Figure 2 compares the density of entropy for Tent, SAR and UniEnt on CIFAR-10-C and SVHN-C (and Figure 6 in the appendix compares OSTTA and SoTTA). Ideally the OOD data would exhibit higher entropy than the InD data to enable separation between the two. However, for most methods, the OOD density does not significantly outweigh the InD density at high entropy. UniEnt achieves the best separation between InD and OOD with entropy confidence.

Figure 5 in the appendix compares the density of max probability on CIFAR-10-C and SVHN-C. The density of max probability for InD is similar across methods with high confidence overall. However, the

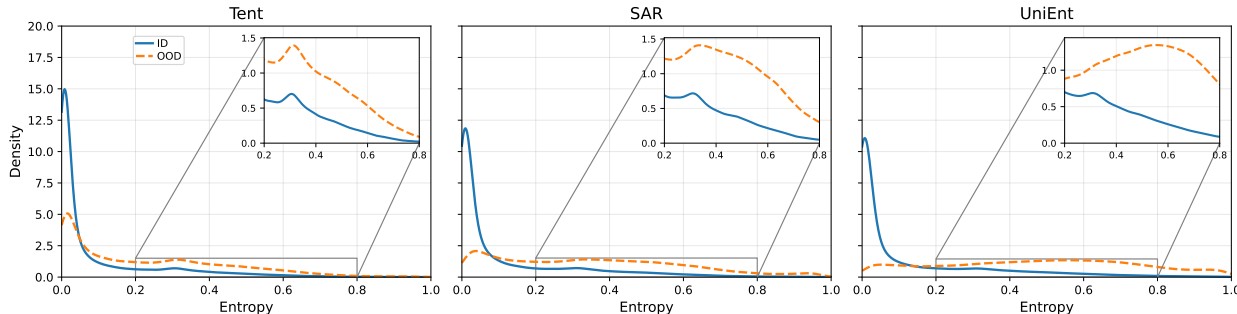

Figure 2: Density of entropy for InD (CIFAR-10-C) and OOD (SVHN-C) data. The InD densities are all similar, but UniEnt differs in its OOD density, perhaps due to its opposing updates on InD vs. OOD data.

Table 5: Performance of open-set TTA on CIFAR-10-C (InD) and CIFAR-100-C (OOD)

| Method | Acc↑ | AUROC↑ | FPR↓ | OSCR↑ |
|---|---|---|---|---|
| Source | 81.74 | 77.61 | 77.22 | 68.26 |
| Source(Sigmoid) | 81.49 | 78.39 | 78.14 | 69.32 |
| Tent | 87.23 | 77.02 | 75.08 | 70.61 |
| Tent(Sigmoid) | 86.05 | 81.43 | 74.89 | 74.40 |
| UniEnt(Tent) | 85.61 | 81.50 | 79.39 | 73.98 |
| UniEnt+(Tent) | 85.51 | 81.42 | 79.71 | 73.87 |
| OSTTA | 86.71 | 77.61 | 75.01 | 70.84 |
| SoTTA | 86.67 | 77.65 | 78.58 | 71.79 |
| SAR(BN) | 86.77 | 80.25 | 78.05 | 73.56 |
| SAR(BN+Sigmoid) | 86.15 | 81.79 | 73.94 | 75.03 |

OOD densities vary across methods, and SoTTA has the highest max probability (even higher than the Tent baseline). UniEnt achieves the best separation between InD and OOD with max probability confidence.

Figure 4 in the appendix examines how InD, OOD, and mixed accuracy vary with the confidence threshold. We plot the mean accuracies across corruption types for UniEnt as it achieves the top mixed accuracy. For entropy confidence, increasing the threshold from 0 to 1 leads to a monotonic increase in InD accuracy, while both OOD accuracy and mixed accuracy decrease. After threshold = 0.1, the OOD accuracy drops more sharply than the gain in InD accuracy, which results in lower mixed accuracy overall. This highlights a trade-off: tuning for high InD accuracy comes at a cost for OOD accuracy, suggesting the need to balance these metrics based on the application.

Figure 7 in the appendix shows the density of sigmoid entropy for Tent (Sigmoid) and SAR (Sigmoid). Compared to softmax entropy, the sigmoid entropy is overall smaller and more concentrated. In addition, the distributions of InD and OOD entropies are bimodal: the InD inputs dominate the first peak, while the OOD inputs dominate the second.

### 4.4 The Effect of Similarity between InD and OOD Data

Table 5 reports performance on CIFAR-10-C with CIFAR-100-C as the OOD data. All methods show overall improvements in InD accuracy compared to SVHN-C as OOD data. Tent achieves strong InD accuracy, but its OOD performance is relatively weaker. Tent with Sigmoid is comparable to UniEnt in OOD performance.

Tables 6 and 7 report the accuracies for thresholding with CIFAR-100-C as the OOD data. InD accuracy is higher for all except UniEnt(+), while OOD accuracy is lower for all. This may be because CIFAR-100-C is more similar to CIFAR-10-C than SVHN-C, allowing updates to adapt to the shifted OOD data in a way that transfers to or at least does not interfere with the InD data. However, this increased similarity also makes it more difficult to recognize InD vs. OOD inputs, as shown by the lower OOD accuracy.

Table 6: Accuracy of TTA on CIFAR-10-C (InD) and CIFAR-100-C (OOD) with entropy confidence.

| Method | InD Acc ↑ | OOD Acc ↑ | Mixed Acc ↑ |
|---|---|---|---|
| Tent | 74.42(87.23) | 63.91 | 69.17 |
| Tent(Sigmoid) | 66.13(86.05) | 67.11 | 66.62 |
| UniEnt(Tent) | 64.34(85.59) | 84.62 | 74.48 |
| UniEnt+(Tent) | 64.07(85.51) | 84.83 | 74.45 |
| OSTTA | 73.35(86.69) | 65.08 | 69.22 |
| SoTTA | 73.95(86.41) | 63.45 | 68.70 |
| SAR(BN) | 69.39(86.93) | 76.21 | 72.80 |
| SAR(BN+Sigmoid) | 68.27(86.15) | 76.44 | 72.36 |

Table 7: Accuracy of TTA on CIFAR-10-C (InD) and CIFAR-100-C (OOD) with max prob. confidence.

| Method | InD Acc ↑ | OOD Acc ↑ | Mixed Acc ↑ |
|---|---|---|---|
| Tent | 78.20(87.23) | 52.84 | 65.52 |
| Tent(Sigmoid) | 73.33(86.05) | 69.32 | 71.33 |
| UniEnt(Tent) | 70.50(85.61) | 74.56 | 72.53 |
| UniEnt+(Tent) | 70.23(85.52) | 74.48 | 72.36 |
| OSTTA | 77.19(86.71) | 53.72 | 65.46 |
| SoTTA | 73.30(86.87) | 51.55 | 62.43 |
| SAR(BN) | 74.79(86.93) | 64.26 | 69.53 |
| SAR(BN+Sigmoid) | 72.10(86.15) | 68.82 | 70.46 |

In Table 8, we compare the InD accuracy on the closed set with the highest InD accuracy observed under two open-set settings. We find that when CIFAR-100-C is used as the OOD test set (referred to as near OOD), the InD accuracy is comparable to—or even higher than—that on the closed set. In contrast, when SVHN-C is used as the OOD test set (far OOD), the InD accuracy drops more. This shows that OOD data that is closer to the training set can cause less drop, and can even help adaptation in some cases.

In Table 9, we do the same near/far comparison at ImageNet scale. In this case, ImageNet-O(-C) serves as the near OOD data, because of its similar object recognition focus, while Textures(-C) serves as the far OOD data, because of its different texture recognition focus. At this scale the near OOD data causes a smaller drop than the far OOD data, but neither aids model performance.

## 4.5 OOD Proportion in Batches

Beyond our controlled experiments in existing settings, we also investigate more realistic deployments. In real settings, the proportion of OOD data in each batch could change. In extreme cases, an entire batch may even consist exclusively of OOD inputs. To account for such variability, we design additional experiments aimed at providing a more comprehensive assessment of these methods. Furthermore, we do not reset the model when the corruption type changes, making the evaluation reflect changing shifts during test time.

In the previous experiments, the proportion of OOD inputs was fixed at 50% across all batches as done in previous benchmarking (Gao et al., 2024; Gong et al., 2023; Lee et al., 2023). However, in real-world settings the test data may not be perfectly balanced, and the OOD proportion may vary substantially. To examine whether the methods remain robust to shift across different proportions, we experimentally compare three OOD proportions: 25%, 50%, and 75%.

Tables 10 report the results under OOD proportions of 25%, 50%, and 75%, respectively. As the proportion of OOD input within a batch increases, both classification accuracy and all OOD detection metrics consistently

Table 8: Best InD Accuracy for CIFAR-10-C closed set and open-set near OOD (CIFAR-100-C) vs. far OOD (SVHN-C). Near OOD data results in a smaller drop or can even improve accuracy.

| Method | Closed Set ↑ | Near OOD ↑ | Far OOD ↑ |
|---|---|---|---|
| Tent | 87.93 | 87.23 | 85.27 |
| Tent(Sigmoid) | 87.14 | 86.05 | 85.08 |
| UniEnt(Tent) | 83.84 | 85.61 | 84.65 |
| UniEnt+(Tent) | 84.74 | 85.52 | 84.44 |
| OSTTA | 87.69 | 86.71 | 85.11 |
| SoTTA | 87.68 | 86.87 | 86.26 |
| SAR(BN) | 86.50 | 86.93 | 85.49 |
| SAR(BN+Sigmoid) | 86.23 | 86.15 | 84.92 |

Table 9: Best InD Accuracy for ImageNet-C closed set and open-set near OOD (ImageNet-O-C) vs. far OOD (Texture-C). Near OOD data results in a smaller drop in accuracy.

| Method | Closed Set ↑ | Near OOD ↑ | Far OOD ↑ |
|---|---|---|---|
| Tent | 9.64 | 5.61 | 2.46 |
| Tent(Sigmoid) | 9.94 | 7.03 | 7.53 |
| UniEnt(Tent) | 37.30 | 35.48 | 34.36 |
| UniEnt+(Tent) | 37.75 | 36.57 | 34.42 |
| OSTTA | 40.61 | 38.30 | 35.24 |
| SoTTA | 6.78 | 3.92 | 2.23 |
| SAR(BN) | 31.67 | 31.03 | 29.80 |
| SAR(BN+Sigmoid) | 26.22 | 25.73 | 23.77 |
| SAR(GN) | 31.44 | 31.43 | 31.40 |

degrade. This indicates that the performance of existing methods is highly sensitive to the OOD ratio in test batches, suggesting that robustness to varying OOD proportions remains an open challenge.

### 4.6 OOD Interval between Batches

In addition to varying the overall proportion of OOD data, we further examined a more realistic setting where OOD inputs appear in intervals across batches. Unlike the proportion experiments, which assume each batch contains a fixed ratio of OOD inputs, the interval setting introduces entire stretches of OOD-only batches between InD batches. This design may better reflect real-world data, where OOD inputs can arrive in bursts rather than being evenly mixed with InD data.

Table 11 presents results under OOD batch intervals of 1, 2 and 3, respectively, when applied in the interval setting. Compared with the fixed-proportion scenario, we find that as the intervals of OOD batches become longer, both accuracy and OOD detection metrics exhibit only minor declines and remain relatively stable.

To gain deeper insight into adaptation updates in the interval setting, we examined the loss dynamics during the first OOD pass with interval size set to 2. The results reveal that, as adaptation proceeds, the OOD loss also decreases, implying that the model's confidence continues to increase even on OOD inputs with incorrect labels. Specifically, the OOD loss delta has a mean of -0.0182 (std = 0.1709), whereas the InD loss delta has a mean of -0.0035 (std = 0.2602). This indicates that OOD inputs generally incur larger losses than InD inputs. Moreover, the smaller variance observed for OOD suggests that their loss distribution is more concentrated, while InD losses are more dispersed. Together, these findings highlight that adaptation may inadvertently reinforce overconfidence on OOD inputs, thereby exacerbating the challenge of distinguishing them from InD inputs.

Table 10: OOD batch proportions performance on ImageNet-C (InD) and ImageNet-O-C (OOD). We evaluate 25%, 50%, and 75% proportion of OOD data in a batch.

| Method | Acc ↑ 25%/50%/75% | | | AUROC ↑ 25%/50%/75% | | | FPR ↓ 25%/50%/75% | | | OSCR ↑ 25%/50%/75% | | |
|---|---|---|---|---|---|---|---|---|---|---|---|---|
| Source | 18.2 | 18.2 | 18.2 | 43.5 | 43.5 | 43.5 | 95.8 | 95.8 | 95.8 | 12.2 | 12.0 | 12.0 |
| Source(Sigmoid) | 15.1 | 15.1 | 15.1 | 46.0 | 46.0 | 46.0 | 95.3 | 95.3 | 95.3 | 9.8 | 9.8 | 9.8 |
| Tent | 7.2 | 5.6 | 4.0 | 45.2 | 42.7 | 39.2 | 97.0 | 97.2 | 98.2 | 4.3 | 3.0 | 1.8 |
| Tent(Sigmoid) | 8.3 | 7.0 | 5.8 | 45.0 | 45.2 | 45.1 | 96.4 | 96.5 | 96.6 | 4.9 | 3.9 | 3.2 |
| UniEnt(Tent) | 36.7 | 35.5 | 32.4 | 47.2 | 50.1 | 53.6 | 96.9 | 95.7 | 92.6 | 23.9 | 24.1 | 22.8 |
| UniEnt+(Tent) | 37.5 | 36.6 | 34.5 | 46.0 | 47.1 | 48.1 | 97.1 | 96.7 | 95.8 | 24.2 | 23.6 | 22.6 |
| OSTTA | 40.1 | 38.3 | 35.0 | 43.8 | 43.0 | 41.9 | 97.6 | 97.6 | 97.4 | 23.6 | 23.0 | 20.8 |
| SoTTA | 4.8 | 3.9 | 2.7 | 47.0 | 47.1 | 42.2 | 95.4 | 95.4 | 97.4 | 3.0 | 2.1 | 1.3 |
| SAR(BN) | 29.6 | 26.3 | 18.4 | 41.4 | 41.0 | 43.8 | 97.3 | 97.1 | 87.2 | 17.9 | 14.4 | 13.5 |
| SAR(BN+Sigmoid) | 26.2 | 25.7 | 24.9 | 47.2 | 47.8 | 48.2 | 96.0 | 95.9 | 95.7 | 16.8 | 16.4 | 16.0 |
| SAR(GN) | 35.3 | 30.2 | 28.7 | 51.3 | 51.7 | 52.9 | 97.7 | 97.3 | 96.8 | 22.9 | 19.8 | 18.0 |

Table 11: OOD batch intervals performance on ImageNet-C (InD) and ImageNet-O-C (OOD). We evaluate 1, 2, and 3 batches of OOD data between each batch of InD data.

| Method | Acc ↑ 1/2/3 Intervals | | | AUROC ↑ 1/2/3 Intervals | | | FPR ↓ 1/2/3 Intervals | | | OSCR ↑ 1/2/3 Intervals | | |
|---|---|---|---|---|---|---|---|---|---|---|---|---|
| Tent | 5.7 | 4.9 | 4.8 | 38.5 | 36.6 | 34.5 | 98.5 | 98.9 | 99.3 | 2.8 | 2.1 | 1.9 |
| Tent(Sigmoid) | 8.4 | 7.9 | 7.7 | 42.7 | 43.3 | 43.6 | 96.8 | 96.8 | 96.7 | 4.7 | 4.5 | 4.4 |
| UniEnt(Tent) | 35.7 | 34.1 | 32.5 | 50.3 | 50.4 | 50.6 | 96.7 | 95.7 | 95.0 | 23.8 | 22.5 | 21.4 |
| UniEnt+(Tent) | 36.7 | 35.7 | 34.8 | 49.2 | 48.0 | 47.2 | 96.7 | 95.5 | 96.3 | 22.5 | 23.1 | 22.1 |
| OSTTA | 37.8 | 36.1 | 35.1 | 44.1 | 40.6 | 38.8 | 97.4 | 97.7 | 97.8 | 23.0 | 20.5 | 19.3 |
| SoTTA | 3.6 | 2.8 | 2.6 | 44.9 | 45.6 | 43.0 | 96.2 | 96.2 | 97.1 | 2.0 | 1.4 | 1.3 |
| SAR(BN) | 13.9 | 21.4 | 19.2 | 40.4 | 38.9 | 36.7 | 97.1 | 97.0 | 97.0 | 7.6 | 11.1 | 9.5 |
| SAR(BN+Sigmoid) | 26.1 | 26.2 | 26.1 | 47.5 | 47.5 | 47.5 | 95.4 | 95.4 | 95.4 | 17.1 | 17.1 | 17.1 |
| SAR(GN) | 31.4 | 31.4 | 31.4 | 47.6 | 47.6 | 47.6 | 97.3 | 97.3 | 97.3 | 19.9 | 19.9 | 19.9 |

This suggests that existing methods are highly sensitive to mixed InD–OOD batches, but largely insensitive to non-mixed scenarios, even when multiple OOD-only batches occur consecutively between InD batches. This observation raises the question of whether the insensitivity is due to the (batch) normalization or the optimization updates by the adaptation methods.

## 4.7 The Effect of Sigmoid Output

Across all settings, Tent (Sigmoid) consistently achieves significantly better OOD metrics compared to the original Tent. The accuracy loss compared to Tent remains below 1% in most cases, except under the 50% OOD proportion open-set setting. In other realistic settings, when there is a change in the OOD data proportion (Tab. 10) or interval (Tab. 11), Tent (Sigmoid) even surpasses the original Tent in accuracy. These results demonstrate that replacing softmax with sigmoid achieves a different InD/OOD trade-off with a clear improvement in the OOD metrics. Although SAR (Sigmoid) typically underperforms the standard softmax-based SAR when the OOD proportion varies, it consistently achieves higher accuracy in the interval setting. Its performance remains stable even as the number of OOD intervals increases.

Overall, sigmoid output with sigmoid entropy for adaptation improves robustness to shift. When the OOD proportion changes, its accuracy drops—as expected—but the drop is substantially smaller than it is for other methods. Likewise, when the number of intervals increases, its accuracy remains stable. These results suggest that sigmoid entropy provides a more reliable loss for adaptation across different proportions and

Table 12: Performance on the alternative shifts of ImageNet-R with open-set data from ImageNet-O.

| Method | Acc | AUROC | FPR | OSCR |
|---|---|---|---|---|
| Tent | 40.96 | 51.41 | 95.21 | 29.23 |
| Tent(Sigmoid) | 33.29 | 56.61 | 86.26 | 24.44 |
| EATA | 43.76 | 50.78 | 97.53 | 31.21 |
| UniEnt(Tent) | 39.45 | 39.16 | 99.20 | 24.46 |
| UniEnt+(Tent) | 40.05 | 43.04 | 98.26 | 26.07 |
| SoTTA | 38.77 | 78.01 | 34.40 | 31.76 |
| OSTTA | 41.33 | 50.86 | 95.81 | 29.36 |
| SAR(BN) | 39.48 | 45.96 | 96.66 | 26.42 |
| SAR(BN+Sigmoid) | 33.66 | 57.03 | 85.80 | 24.83 |
| SAR(GN) | 40.81 | 26.27 | 99.65 | 18.63 |

Table 13: Sensitivity analysis for SAR: we examine the average performance under different OOD proportions with different normalization layers on ImageNet-C (InD) and ImageNet-O-C (OOD).

| OOD Prop. | Arch. | Layer | Acc↑ | AUROC↑ | FPR↓ | OSCR↑ |
|---|---|---|---|---|---|---|
| 25% | ResNet | BN | 29.63 | 41.41 | 97.29 | 17.93 |
| 50% | ResNet | BN | 26.30 | 41.05 | 97.12 | 14.35 |
| 75% | ResNet | BN | 18.42 | 43.78 | 87.22 | 13.52 |
| 25% | ResNet | GN | 35.26 | 51.25 | 97.72 | 22.90 |
| 50% | ResNet | GN | 30.22 | 51.67 | 97.28 | 19.76 |
| 75% | ResNet | GN | 28.67 | 52.86 | 96.80 | 17.94 |
| 25% | ViT | LN | 54.44 | 69.25 | 85.77 | 44.15 |
| 50% | ViT | LN | 53.73 | 68.51 | 86.93 | 43.60 |
| 75% | ViT | LN | 55.58 | 70.95 | 84.97 | 38.61 |

rates of InD and OOD data. Note however that sigmoid output can involve a trade-off: the improvement in OOD metrics comes at a cost in InD accuracy in multiple cases.

### 4.8 More Shifts: Renditions

ImageNet-R (Hendrycks et al., 2021a) for "Rendition" provides more shifts to evaluate adaptation. It includes 200 of the 1000 ImageNet classes in different artistic styles, or renditions, which cover cartoons, graffiti, paintings, sculptures, toys, and more. These diverse variations differ in appearance from the ImageNet training data and from the synthetic image corruptions of ImageNet-C.

For open-set evaluation we pair ImageNet-R with ImageNet-O. We convert the ImageNet source models, for 1000 classes, to the ImageNet-R models, for 200 classes, following the standard practice to predict only the ImageNet-R classes: select the logits for these classes then normalize by the softmax. We report accuracy and the open-set metrics in Table 12.

### 4.9 Normalization Layers and Open-Set Adaptation

Normalization layers are ubiquitous in deep networks and are a common choice for test-time updates. Tent for instance updates only the parameters of batch normalization layers while updating their statistics on the test data at the same time. Further experiments for SAR investigate the impact of different normalization layers, and find the choice of normalization matters for closed-set adaptation. To complement their analysis we check the effect of different normalization layers on open-set adaptation.

We compare ResNet-50 with batch normalization (BN), ResNet-50 with group normalization (GN), and ViT-B/16 with layer normalization (LN) to cover different types of normalization and architecture. Table 13 reports the results under varying OOD proportions, showing the effects of each normalization layer on

accuracy and OOD metrics. The results indicate that BN exhibits large fluctuations in both Acc and FPR@TPR95, GN shows relatively smaller fluctuations in accuracy, while LN demonstrates the most stable performance overall.

## 5 Conclusion: More Updates are Needed to Balance InD and OOD

We benchmark several recent robust and open-set TTA methods to measure InD and OOD accuracy. Our results show that methods can achieve strong InD accuracy, but at the cost of OOD accuracy, highlighting a current conflict between robust adaptation on InD data and reliable rejection of OOD data. Our systematic evaluation provides further evidence of this across methods, shifts, and the choice of open-set data.

Furthermore, we evaluate the ability of different confidence scores to separate InD and OOD inputs by simple thresholding. These thresholds help to measure the effectiveness of the confidence-based filtering in SAR and SoTTA and the more sophisticated filtering in UniEnt(+) and OSTTA. Entropy generally provides better OOD accuracy than maximum probability across methods, as measured by the softmax response, suggesting its potential as a more reliable score for open-set adaptation.

In addition, we test a more realistic setting where the proportion of InD and OOD inputs varies across batches from 0% to 100%. The effect of the proportion depends on the choice of normalization layer (BN, GN, or LN). For models with BN, performance is much more sensitive to the batch mixture setting, with InD and OOD data in each batch, than to the batch interval setting, with InD and OOD data alternating across batches. In these conditions, GN is the most stable, while BN is the least stable.

Our results highlight the opportunity for open-set TTA to better balance InD adaptation and OOD rejection to mitigate errors on known and unknown classes during deployment. Our analysis of confidence on InD vs. OOD data suggests the potential for better filtering to improve OOD accuracy. Finally, our new baseline of entropy minimization with sigmoid output, which we have instantiated for Tent and SAR, reveals another direction for balancing InD and OOD accuracy: changing the output distribution to better separate InD and OOD data without additional filtering. While we examine the effect of sigmoid output on InD accuracy and OOD rejection, we have not investigated other potential effects of the output distribution, such as calibration (Guo et al., 2017). As the gain in OOD metrics for sigmoid output is a trade-off, which sacrifices InD accuracy, other output distributions or updates may be needed for strict improvement.

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

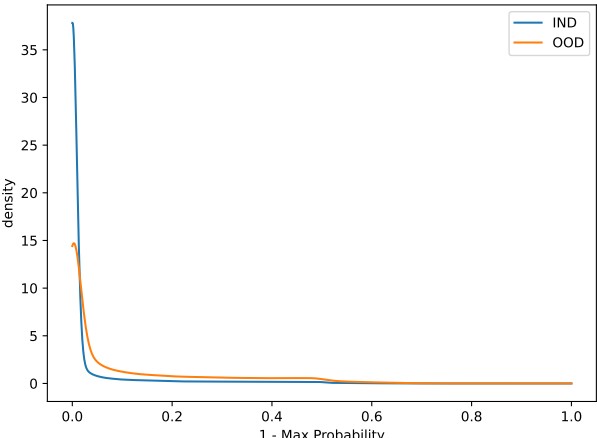

Figure 3: Density of the max probability on CIFAR-10-C (InD) and SVHN-C (OOD) data of Tent(Sigmoid).

Kate Saenko, Brian Kulis, Mario Fritz, and Trevor Darrell. Adapting visual category models to new domains. In *European conference on computer vision*, pp. 213–226. Springer, 2010.

Kuniaki Saito, Shohei Yamamoto, Yoshitaka Ushiku, and Tatsuya Harada. Open set domain adaptation by backpropagation. In *Proceedings of the European conference on computer vision (ECCV)*, pp. 153–168, 2018.

Steffen Schneider, Evgenia Rusak, Luisa Eck, Oliver Bringmann, Wieland Brendel, and Matthias Bethge. Improving robustness against common corruptions by covariate shift adaptation. *Advances in neural information processing systems*, 33:11539–11551, 2020.

Yu Sun, Xiaolong Wang, Zhuang Liu, John Miller, Alexei Efros, and Moritz Hardt. Test-time training with self-supervision for generalization under distribution shifts. In *International conference on machine learning*, pp. 9229–9248. PMLR, 2020.

Dequan Wang, Evan Shelhamer, Shaoteng Liu, Bruno A. Olshausen, and Trevor Darrell. Tent: Fully test-time adaptation by entropy minimization. In *International Conference on Learning Representations*, 2021.

Kaichao You, Mingsheng Long, Zhangjie Cao, Jianmin Wang, and Michael I Jordan. Universal domain adaptation. In *Proceedings of the IEEE/CVF conference on computer vision and pattern recognition*, pp. 2720–2729, 2019.

Sergey Zagoruyko and Nikos Komodakis. Wide residual networks. In *BMVC*, 2016.

# A Appendix

Fig. 3 shows the distribution of max_prob for Tent with sigmoid. Compared with Fig. 7, max_prob provides a cleaner separation between InD and OOD inputs with a single threshold, while the entropy distribution becomes more bimodal and overlapping after Tent (sigmoid), making entropy thresholding less reliable in this setting.

# B Supplementary Plots

Figures 4a and 4b show the the InD, OOD, and mixed accuracies of UniEnt on CIFAR-10-C and SVHN-C as a function of confidence threshold for entropy and max probability respectively.

Figure 5 shows the density of 1 - max_probability for the five methods on CIFAR-10-C and SVHN-C.

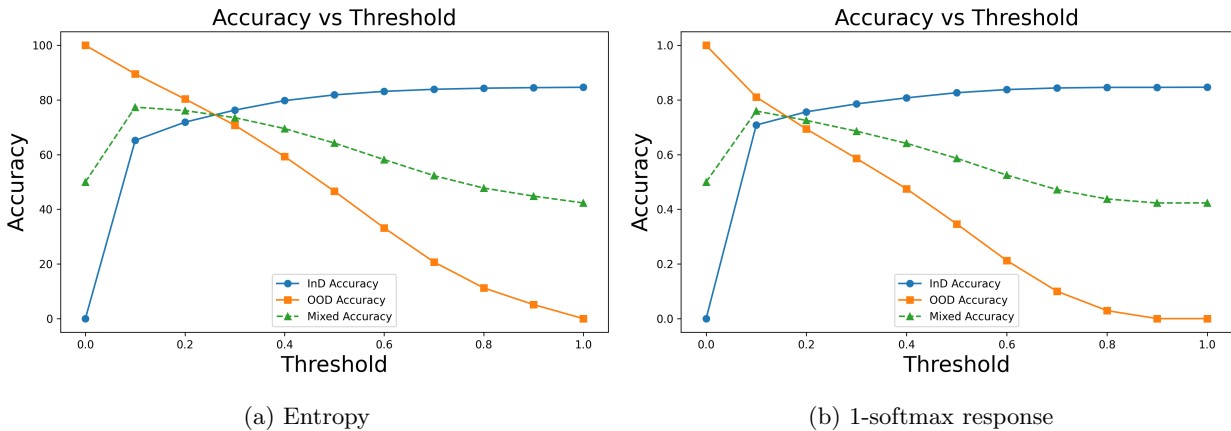

(a) Entropy                                          (b) 1-softmax response

Figure 4: Accuracy for InD, OOD, and their average (Mixed) on CIFAR-10-C (InD) and SVHN (OOD) across confidence thresholds for UniEnt. We plot the mean accuracies across all 15 corruption types. InD and OOD accuracy trade off: the mixed accuracy can balance the two and improve up to a point, but then higher thresholds result in overall lower mixed accuracy.

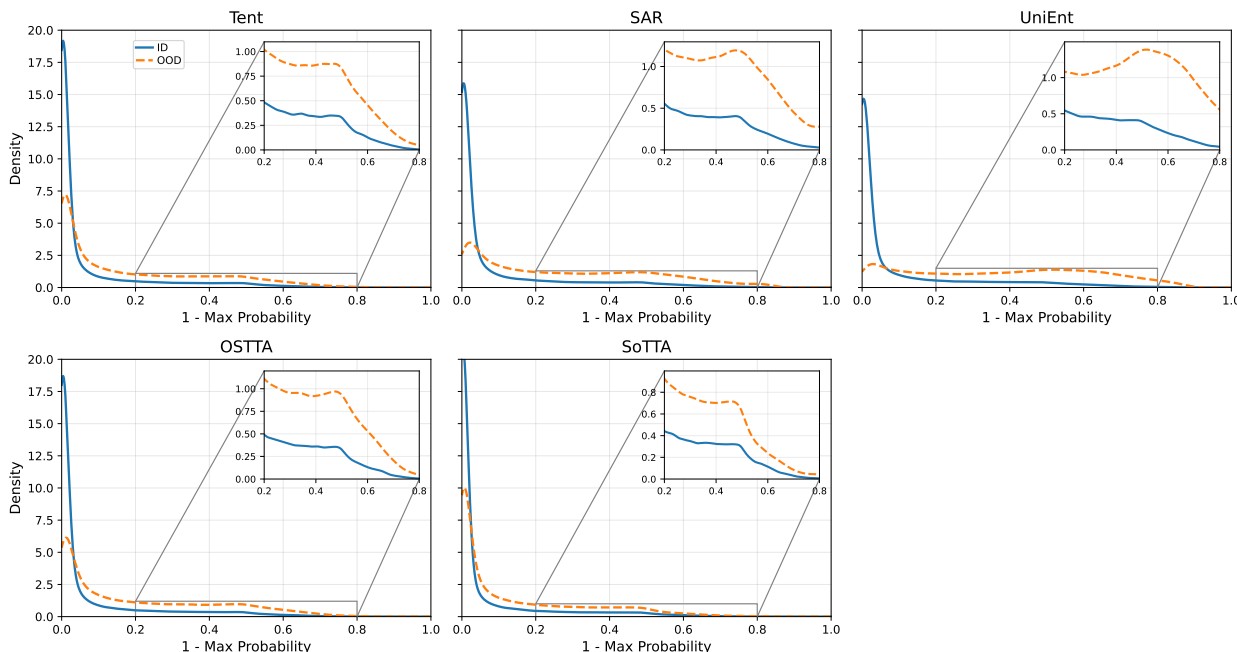

Figure 5: Density of 1 - max probability on CIFAR-10-C (InD) and SVHN-C (OOD) data.

Figure 6 shows the density of entropy for OSTTA and SoTTA on CIFAR-10-C and SVHN-C.

Figure 7 shows the density of entropy for Tent (Sigmoid) and SAR (Sigmoid) on CIFAR-10-C and SVHN-C.

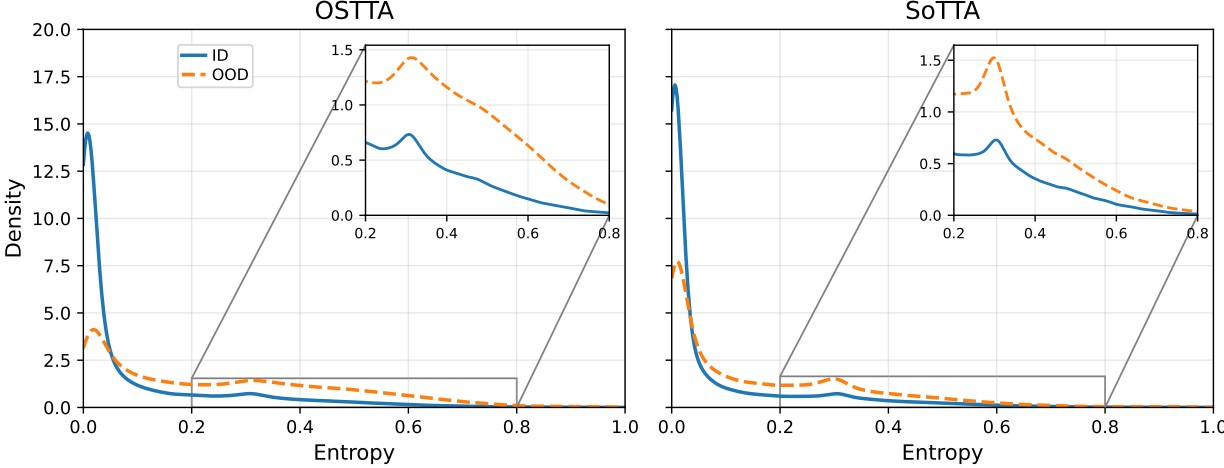

Figure 6: Density of entropy on CIFAR-10-C (InD) and SVHN-C (OOD) data. See Figure 2 for the other open-set TTA methods.

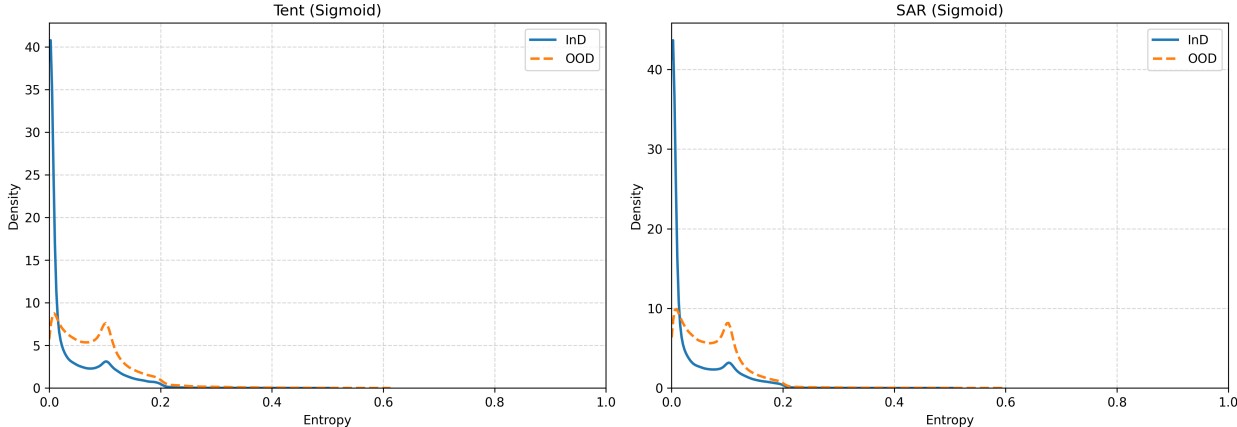

Figure 7: Density of the sigmoid entropy on CIFAR-10-C (InD) and SVHN-C (OOD) data. Compare the difference for the softmax editions of Tent and SAR in Figure 2.

