# OpenReview forum: "A Closer Look at In-Distribution vs. Out-of-Distribution Accuracy for Open-Set Test-time Adaptation"
_TMLR — Accepted by TMLR_

### Review · Reviewer_4RwD · 2026-01-11

**Summary Of Contributions:**

Summary
The paper benchmarks existing robust and Open-Set Test-Time Adaptation (TTA) methods (SAR, SoTTA, OSTTA, UniEnt) under the presence of Out-of-Distribution (OOD) data. It highlights a trade-off where current methods often prioritize In-Distribution (InD) accuracy at the expense of OOD detection capability. The authors propose a "new" baseline that replaces the standard softmax output with a sigmoid-based multi-label output to better handle rejection of unknown classes. Additionally, the study evaluates these methods under more realistic conditions, such as varying proportions of OOD data and OOD intervals.

Strengths
1. The paper moves beyond the standard fixed-ratio assumptions in Open-Set TTA. The experiments detailed in Section 4.5 (varying OOD proportions) and Section 4.6 (OOD intervals) provide insights into how methods fail when the test stream is not perfectly balanced (50/50).
2. The work covers a reasonable range of datasets (CIFAR-10-C/SVHN, ImageNet-C/ImageNet-O/Texture) and evaluates multiple methods (UniEnt, OSTTA, SoTTA, SAR).
3. Figure 3 (Appendix) effectively visualize the inherent tension between adapting to shifted InD data and rejecting OOD samples.

Weaknesses
1. The paper argues in Section 4.3 that "entropy generally provides better OOD accuracy." However, Table 3 (Max Prob confidence) shows Tent(Sigmoid) performing much better (74.78%) than in Table 2 (Entropy confidence, 61.41%). This creates confusion: if entropy is better, why is this gap so large? A discussion on this could strengthen the paper.

2. Figure 2 and Figure 6 (density plots) are somewhat difficult to read due to the overlapping distributions and small scales. Besides, the figures do not seem to be vector graphics; please consider replacing them with vector formats (e.g., PDF or EPS) for better clarity.
The paper mentions tuning thresholds to "maximize the mixed accuracy" (Section 4). In a more realistic test-time adaptation setting, we do not have labels to tune this threshold on the fly. This assumes an "oracle" validation set, which slightly undermines the "Test-Time" claim.

3. Section 5 discusses normalization layers (BN vs GN vs LN) briefly (Table 12). While interesting, this section feels less connected from the main narrative of InD/OOD trade-off and Sigmoid outputs. It might be more suitable for an appendix section.

4. The writing quality of the paper requires improvement. A considerable portion of the text is difficult to follow and occasionally confusing, giving the impression that the paper may have undergone an imperfect translation process. Specific examples from the text include: "This design may better reflects real-world data streams."; "ImageNet-O is nearer to ImageNet" and "Textures is farther from ImageNet."; Phrases such as "update to try to maintain accuracy" are wordy and unnatural. I recommend that the authors perform a comprehensive polishing which would greatly enhance readability.

**Additional Comments:**

N/A

**Audience:**

Yes

**Audience Explanation:**

The paper benchmarks existing robust and Open-Set Test-Time Adaptation (TTA) methods (SAR, SoTTA, OSTTA, UniEnt) under the presence of Out-of-Distribution (OOD) data.  That would allign the topic in TMLR

**Broader Impact Concerns:**

The paper can strengthen the broader concerns about data robustness and model generalization.

**Claims And Evidence:**

Yes

**Claims Explanation:**

On the trade-off between InD and OOD performance: Tables 1, 2, and 3 clearly demonstrate an inverse relationship between InD accuracy and OOD accuracy. The data indicates that thresholds optimized for InD performance result in OOD recognition rates approaching zero.
The paper compares the original versions of Tent/SAR with their Sigmoid counterparts across multiple tables (e.g., Tables 1, 10, and 11). The results consistently show that the Sigmoid variants generally outperform Softmax versions in terms of AUROC and FPR metrics, while demonstrating greater robustness when handling varying OOD ratios.

On the sensitivity to realistic scenarios (ratios and intervals): Table 10 illustrates a sharp decline in the performance of BN-based methods as the OOD ratio increases from 25% to 75%.
The authors utilize community-recognized datasets (CIFAR-10-C, ImageNet-C) and standard OOD sources (SVHN, ImageNet-O, etc.).

**Requested Changes:**

Please see the weakness part above

---

> ### Author Response · Authors · 2026-03-04
>
> 1. The paper argues [...] entropy generally provides better OOD accuracy This creates confusion: if entropy is better, why is this gap so large? A discussion on this could strengthen the paper.
>
> Thank you for raising this point. In most cases, entropy provides better OOD accuracy, as seen in Table 2 vs. Table 3, with the exception of Tent with sigmoid. To better understand this case, we plot the distributions of entropy and max_prob for Tent with sigmoid. In these plots, max_prob provides a cleaner separation between InD and OOD samples with a single threshold, whereas the entropy distribution becomes more bimodal and overlapping after Tent (sigmoid), making entropy thresholding less reliable in this setting.
>
> 2. Figure 2 and Figure 6 (density plots) are somewhat difficult to read due to the overlapping distributions and small scales.
>
> Thank you for the suggestion. We have changed these plots to vector format to improve legibility. In the appendix, some plots are shown two at a time and side-by-side. If this arrangement is easier to read, we can adopt this formatting for more of the results to give each plot additional space.
>
> 3. Section 5 discusses normalization layers (BN vs GN vs LN) briefly (Table 12)
>
> We thank the reviewer for pointing this out. In this section we examine the effect of normalization layers on open-set data to complement the analysis of SAR on closed-set data. In the revision, we move this discussion to Sec. 4.9 and edit the text to more clearly explain the purpose of these experiments.
>
> 4. The writing quality of the paper requires improvement
>
> We thank the reviewer for the close reading. In the revision, we have further edited the paper to reduce wordiness and improve the overall clarity and readability.

---

> > ### Comment · Reviewer_4RwD · 2026-03-26
> > **Responses**
> >
> > Thank you for the detailed response. My major concerns are adequately addressed. One remaining area for improvement is to adjust the layout of certain figures and tables to make the paper more compact.

---

### Review · Reviewer_cYv6 · 2026-02-15

**Summary Of Contributions:**

The paper studies open-set test-time adaptation and argues that existing approaches often improve in-distribution accuracy while remaining vulnerable to out-of-distribution samples during adaptation. It proposes a unified evaluation protocol that jointly measures classification and rejection performance under mixed test streams, and conducts extensive empirical analysis across several datasets. The paper also explores simple design choices, such as replacing softmax outputs with sigmoid predictions, and shows that these changes can improve robustness in open-set scenarios.

Strengths include the practical relevance of the problem, systematic experimentation, and a useful evaluation perspective. Weaknesses include limited methodological novelty, since the main technical modification is relatively simple and similar ideas have appeared in related open-set literature.

**Audience:**

Yes

**Audience Explanation:**

Test-time adaptation and robustness under distribution shift are active research topics, and the open-set scenario is particularly relevant for real-world deployment. The evaluation framework and empirical observations in this paper help clarify practical limitations of current approaches. Even though the algorithmic contribution is modest, the diagnostic insights and benchmarking perspective should be useful to researchers working on domain adaptation, robustness, and open-world recognition.

**Broader Impact Concerns:**

I do not see major ethical concerns specific to this work. The paper focuses on robustness evaluation and adaptation methods, which are standard machine learning topics. No additional broader impact discussion appears strictly necessary beyond normal statements on safe deployment of machine learning systems.

**Claims And Evidence:**

Yes

**Claims Explanation:**

The paper provides extensive empirical results across multiple datasets, corruption settings, and open-set configurations. The experiments consistently demonstrate the trade-off between in-distribution accuracy and OOD rejection, and the reported improvements from the proposed design choices are supported by quantitative comparisons. While the main methodological idea is simple, the experimental evidence itself appears clear and generally convincing.

**Requested Changes:**

Clarify positioning relative to prior open-set filtering methods (critical).
The paper sometimes suggests that earlier methods largely ignore OOD handling, while in practice many approaches already use entropy or confidence filtering. The authors should more clearly acknowledge this and frame their contribution as a systematic evaluation and analysis rather than a correction of a completely overlooked issue.

Better emphasize the main contribution as evaluation and analysis (important).
Since the technical modification itself is simple, the paper would benefit from explicitly highlighting that its primary value lies in the benchmark design, stress testing, and empirical findings.

Add discussion on limitations of the sigmoid replacement (optional but helpful).
For example, discuss possible calibration issues, effects on class competition, or cases where the change may hurt performance.

Minor clarity improvements (optional).
Some sections could more clearly distinguish experimental observations from broader claims about real-world deployment.

---

> ### Author Response · Authors · 2026-03-04
>
> 1. Clarify positioning relative to prior open-set filtering methods
>
> We appreciate the reviewer’s suggestion and agree that this positioning can be made clearer. The role of our work is to provide further benchmarking and a simple new baseline for existing open-set methods, rather than to compete with or replace them. We hope this helps inform the community about gaps in prior evaluations (e.g., open-set metrics that were not consistently reported across methods) and about the performance of current filtering methods relative to simple confidence thresholding.
>
> To clarify this, we revise Sec. 3 to include the following orientation: Existing open-set methods address filtering OOD data and may use confidence measures such as entropy, while our purpose is to more systematically examine their performance. In particular, we evaluate the effect of their filtering and update rules on both InD accuracy and open-set metrics, and compare them to simpler baselines that threshold confidence alone under different output distributions.
>
> 2. Better emphasize the main contribution as evaluation and analysis
>
> Thank you for the feedback on clarifying the main contribution. We agree that our benchmarking and analysis are the main focus for informing the community. The simple sigmoid modification is intended as a counterpart to our analysis of softmax-based confidence measures, rather than as a new method intended to replace existing approaches.
>
> 3. Add discussion on limitations of the sigmoid replacement
>
> We agree that it is important to note limitations and potential side-effects. While we examine the effect of sigmoid outputs on InD accuracy and OOD rejection, we have not investigated other potential effects on the output distribution, such as calibration. In addition, in our experiments the improved OOD metrics with sigmoid can involve a trade-off and may come at the cost of some InD accuracy.

---

> > ### Comment · Reviewer_cYv6 · 2026-03-05
> >
> > Thank you for the response and clarifications. While I appreciate the acknowledgement of the reviewer’s concerns and the intention to revise the paper accordingly, the current reply remains somewhat high-level. In particular, the response mainly states that revisions will be made, but does not provide concrete details on what specific changes (e.g., additional citations, revised wording, or new analyses/experiments) will be incorporated in the manuscript. It would be helpful if the authors could outline more explicitly what modifications will be made to address the positioning relative to prior open-set methods, how the benchmarking contribution will be emphasized in the paper, and whether any additional analysis regarding the sigmoid replacement will be included. Providing more concrete details would help clarify how the concerns will be resolved and will inform my final evaluation.

---

> > > ### Author Response · Authors · 2026-03-05
> > > **Thank You and Please See the Revision (the New PDF)**
> > >
> > > Thank you for your attention and quick response. Along with our response we uploaded a revision of the PDF as enabled by the TMLR process. Sorry for not signalling this better and we will make a top-level comment shortly to better point this out. We would appreciate your review of this revised edition.
> > >
> > > > additional citations, revised wording, or new analyses/experiments)
> > >
> > > Yes, we have added more citations to the introduction to better point to existing work, revised wording in multiple sections (incl. the end of Sec. 1, the beginning of Sec. 3, and in Sec. 5), and added experiments with more shifts (Tab. 12) and tuning to a held-out set of corruptions for more realistic use (Tabs. 2-4 and 6-9 on thresholding) to provide more evaluation of the existing methods and the sigmoid baseline. Please let us know if we can highlight the changes by for instance adding color to content that is edited and added. We can do so if this would help inform the review process.
> > >
> > > Thank you again for the helpful suggestions. We are happy to continue to respond and revise.

---

### Review · Reviewer_qMJS · 2026-02-18

**Summary Of Contributions:**

This paper re-examines open-set test-time adaptation (TTA) through the lens of a joint In-distribution (InD) accuracy vs. OOD rejection accuracy trade-off, arguing that prior evaluations over-emphasize InD accuracy while under-reporting OOD correctness. It benchmarks several representative robust/open-set TTA methods (e.g., SAR, OSTTA, UniEnt, SoTTA) on corruption-based shifts at CIFAR-10-C and ImageNet-C scales, with OOD sets including SVHN-C/CIFAR-100-C and ImageNet-O-C/Textures-C, and reports both accuracy and standard OOD metrics (AUROC, FPR@TPR95, OSCR).

A key additional contribution is a simple baseline that replaces softmax with sigmoid outputs to enable explicit rejection (all class probabilities low), implemented by lightly adapting the final layer on source data and then applying Tent/SAR-style test-time updates.

Strengths:

1.Clear problem framing: highlights a genuine evaluation gap (OOD accuracy often missing) and standardizes metrics/plots to expose the trade-off.
2.Empirically grounded: compares multiple methods and reports both accuracy + OOD metrics; additionally checks the effectiveness of methods’ own OOD filtering.
3.Practical insight: shows that tuning for maximal InD accuracy can collapse OOD accuracy toward zero, motivating deployment-aware reporting.

Weaknesses:

1.The main experimental setting (50/50 InD–OOD mixing with threshold tuned to maximize mixed accuracy) risks being somewhat oracle-like and may not reflect real deployment constraints.
2.Corruption benchmarks are useful but narrow; conclusions may not fully transfer to other shifts (e.g., style/domain shifts, temporal drift) or to more diverse open-set conditions.
3.Some baselines are run under constrained/atypical configurations (e.g., SAR applied with BN despite being designed for GN/LN), which the paper acknowledges but still affects the strength of comparative claims.

**Additional Comments:**

N/A

**Audience:**

Yes

**Audience Explanation:**

Open-set TTA sits at the intersection of robustness, adaptation, and OOD detection, all of which are central to TMLR’s audience. The paper’s main value is not a new complex method, but a careful diagnostic benchmark + conceptual takeaway: optimizing InD performance alone can be misleading, and open-set adaptation pipelines can silently fail at OOD rejection and OOD filtering.

The sigmoid-output baseline is also of practical interest because it is simple and highlights that architectural/probabilistic output choices can materially affect open-set separability, even without explicit OOD modules.

**Broader Impact Concerns:**

I do not see immediate broader-impact red flags (no human data, no sensitive attributes, no direct high-risk deployment claims). The most relevant concern is deployment misuse: if practitioners adopt open-set TTA without careful OOD evaluation, it could increase false acceptances of unknown inputs in safety-critical settings. This is already aligned with the paper’s motivation and can be addressed by a short Broader Impact note emphasizing the importance of reporting OOD metrics and failure modes.

**Claims And Evidence:**

Yes

**Claims Explanation:**

The central claims are primarily empirical (i.e., current open-set TTA methods struggle to balance InD accuracy and OOD rejection; filtering is imperfect; sigmoid outputs alter the trade-off). These claims are supported by: (i) explicit reporting of InD/OOD/mixed accuracies under confidence thresholding, (ii) standard OOD metrics (AUROC/FPR@TPR95/OSCR), and (iii) an additional check of each method’s internal filtering effectiveness.

However, there are two methodological caveats that temper how “deployment-convincing” the evidence is:

1. Thresholds are tuned to maximize mixed accuracy (and the paper also reports best-InD thresholds), but this tuning may implicitly use information that is not readily available online at test time; a more realistic threshold selection protocol would strengthen the conclusions.
2. Some comparisons are not fully “apples-to-apples” w.r.t. recommended model/normalization choices for certain methods (e.g., SAR). The authors acknowledge this, but additional runs following each method’s best practices would make the evidence more convincing.

**Requested Changes:**

1. Current results rely on threshold sweeps to maximize mixed accuracy. Please add a protocol that selects thresholds without access to the full test distribution (e.g., calibration on a small held-out stream, or unsupervised/streaming thresholding), and report sensitivity.

2. Bring method configurations closer to each method’s intended setting, or clearly separate “controlled comparability” experiments from “best-practice performance” experiments. E.g., SAR is acknowledged to be designed for GN/LN; provide primary results following each method’s recommended backbone/normalization (or clearly qualify comparative conclusions).

3. Define the evaluation stream assumptions more explicitly (batch sizes, ordering, whether the corruption type changes, whether the model is reset, etc.) and ensure reproducibility (ideally with code). This is essential because open-set TTA results can be fragile to stream details.

4. Broaden shifts beyond corruption benchmarks (at least one additional shift family) to support claims about open-set TTA behavior more generally.

5. For the sigmoid baseline, include an ablation clarifying the role of the “final-layer fine-tuning” step (how much it contributes vs. sigmoid itself), since the method explicitly modifies training before test-time adaptation.

6. Report compute/latency (esp. for ImageNet-scale) and memory overhead, since TTA methods are often deployment-motivated.

---

> ### Author Response · Authors · 2026-03-04
>
> 1. Please add a protocol that selects thresholds without access to the full test distribution
>
> Thank you for this suggestion to make the evaluation more realistic. For this response and the revision we instead tune the thresholds on the four held-out corruptions (speckle noise, Gaussian blur, spatter, and saturate) rather than the set of fifteen test corruptions. We select these thresholds to achieve either the highest in-distribution accuracy or mixed accuracy per the choice of tuning objective. Across most methods, the chosen thresholds are identical to those chosen with access to the full test sets, and the results across the accuracy and open-set metrics are the same.
> We have edited Sec. 4 to more fully explain the hyperparameter selection and identify the two tuning schemes: the suggested and realistic tuning—on held-out shifts—and our optimistic tuning—on the test shifts—intended as an analysis of the best achievable results for thresholding.
>
> 2. Bring method configurations closer to each method’s intended setting
>
> With the exception of SAR, each method is already evaluated in its intended setting. For SAR, we modify it for consistency with the other methods, but to address this helpful feedback we now (1) better indicate this modification in the tables and (2) we also evaluate the standard SAR configurations of ResNet50 with GN and ViT-B/16 with LN on ImageNet. For CIFAR-10-C, we cannot simply evaluate standard SAR because it relies on GN/LN, which is incompatible with the WideResNet architecture used in these CIFAR-10-C experiments. While ResNet could in principle be used instead, the results would not be directly comparable to the other methods evaluated on WideResNet, and there is no publicly available ResNet checkpoint for CIFAR-10 under the same experimental setting. For ImageNet-C, we evaluate SAR with ResNet50(GN) in Tab. 10, and we evaluate SAR with ResNet50(GN/BN) and ViT-B/16(LN) in Tab 13. Although this differs from the standard ResNet50(BN) model used for the other methods and thus is not strictly comparable, it allows us to include the actual effectiveness of SAR under its suggested architecture for reference.
>
> 3. Define the evaluation stream assumptions more explicitly
>
> We edited Sec. 3.1 to clarify this point (please see the paragraph opening with "Data."). We use a batch size of 200 for CIFAR-10-C experiments and 64 for ImageNet-C experiments. We evaluate the standard sequence of 15 corruption types. For CIFAR-10-C, we reset the model after each corruption, evaluating episodic shift. For ImageNet-C, we do not reset, evaluating continual shift.
>
> 4. Broaden shifts beyond corruption benchmarks
>
> To broaden the evaluation beyond corruption benchmarks, we include the shifts of ImageNet-R paired with open-set data from ImageNet-O in our experiments. This additional dataset provides different artistic renditions of 200 ImageNet classes, including graffiti, sculptures, toys, and other styles, introducing more diverse shifts that complement the common image corruptions in ImageNet-C. The results are shown in Table 12.
>
> 5. For the sigmoid baseline, include an ablation clarifying the role of the “final-layer fine-tuning” step
>
> We include numbers in Sec. 3.2. On CIFAR-10, the original softmax model achieves 93.91% accuracy, while our fine-tuned model achieves 93.90%. The role of this fine-tuning is so that the logits/linear output of the model are compatible with the choice of nonlinearity: the sigmoid vs. the softmax. In contrast, directly replacing the final softmax layer with a sigmoid without fine-tuning leads to a significant drop to 89.28% accuracy, indicating that the fine-tuning step is necessary to restore the model’s performance and ensure a fair comparison starting from similar clean accuracy.
>
> 6. Report compute/latency (esp. for ImageNet-scale) and memory overhead
>
> Thank you for highlighting the importance of computation, which is indeed a key consideration for test-time adaptation. To clarify, the computation in our benchmarking and our sigmoid baseline is essentially unchanged: the choice of softmax vs. sigmoid nonlinearity at the output and the thresholding are not significant in time or memory relative to the computation of the predictions and the optimization updates.
> On ImageNet-C, our timing results show that the original Tent and Tent (sigmoid) both take about 175 ms per batch, so the sigmoid replacement does not add noticeable runtime overhead. The peak GPU memory increases only slightly from 5533.2 MB to 5633.2 MB (about 100 MB, <2%), indicating only a marginal increase in memory usage. We have added this information at the end of Section 3.2.
> If it would make the work more self-contained, we can also include the computation time and peak memory usage for each method on ImageNet. Could the reviewer confirm whether these are the most relevant measurements?

---

### Author Response · Authors · 2026-03-05
**Please See the Edits in the Revision (the New PDF)**

We thank all of the reviewers for their constructive and detailed feedback. Along with our responses we have also uploaded a revision as a new version of the submission PDF. When we refer to sections, tables, and figures in the reply we are referring to the revision. To respect reviewer time we have tried to summarize the changes in our responses, and so we offer the revision in case the full edits can provide more detail for discussion and the final evaluation.

---

### Decision · Action_Editor_UxgV · 2026-03-27

**Recommendation:** Accept with minor revision

**Audience:**

Yes

**Audience Explanation:**

OOD generalization is an important topic in the ML community. The authors provide their insights through empirical experimental results.

**Claims And Evidence:**

Yes

**Claims Explanation:**

The discussions are valuable. The authors have adequately addressed the concerns raised by the reviewers. The current presentation is suitable for publication, but all remaining issues should be fully revised in the final version.